# Design of GHz Mechanical Nanoresonator with High *Q*-Factor Based on Optomechanical System

**DOI:** 10.3390/mi13111862

**Published:** 2022-10-30

**Authors:** Jun Jin, Ningdong Hu, Lamin Zhan, Xiaohong Wang, Zenglei Zhang, Hongping Hu

**Affiliations:** 1Department of Mechanics, School of Aerospace Engineering, Huazhong University of Science and Technology, Wuhan 430074, China; 2Hubei Key Laboratory of Engineering Structural Analysis and Safety Assessment, Huazhong University of Science and Technology, Wuhan 430074, China; 3School of Optical and Electronic Information, Huazhong University of Science and Technology, Wuhan 430074, China; 4Wuhan Second Ship Design and Research Institute, Wuhan 430074, China

**Keywords:** nanoresonator, GHz, *Q*-factor, optomechanical system, NEMS

## Abstract

Micro-electromechanical systems (MEMS) have dominated the interests of the industry due to its microminiaturization and high frequency for the past few decades. With the rapid development of various radio frequency (RF) systems, such as 5G mobile telecommunications, satellite, and other wireless communication, this research has focused on a high frequency resonator with high quality. However, the resonator based on an inverse piezoelectric effect has met with a bottleneck in high frequency because of the low quality factor. Here, we propose a resonator based on optomechanical interaction (i.e., acoustic-optic coupling). A picosecond laser can excite resonance by radiation pressure. The design idea and the optimization of the resonator are given. Finally, with comprehensive consideration of mechanical losses at room temperature, the resonator can reach a high *Q*-factor of 1.17 × 10^4^ when operating at 5.69 GHz. This work provides a new concept in the design of NEMS mechanical resonators with a large frequency and high *Q*-factor.

## 1. Introduction

The progress of nanotechnology has promoted the rapid development of nano-electromechanical systems (NEMS) in recent years. Their high frequency, low damping, i.e., high mechanical quality factor (*Q*-factor) and small mass make them central components for next-generation clocks, filters, resonators, sensors, and quantum technologies. Based on the inverse piezoelectric effect, the quartz resonator [1,2,3,4], surface acoustic wave (SAW) device and interdigital transducer (IDT) [5] can be applied for resonators from kilohertz to megahertz. Moreover, some special SAW devices can also operate in GHz [6]. To meet the needs of today’s wireless technology, film bulk acoustic resonators (FBAR) are utilized to generate GHz resonance and show excellent performance in temperature stability [7,8]. However, the low quality factor becomes a main obstacle that limits the further development of both SAW and FBAR. Moreover, the existing techniques often require piezoelectric materials with an external radiofrequency excitation, which are not readily integrated into existing CMOS infrastructures, while non-piezoelectric structures are very inefficient [9]. In recent years, optomechanical systems have seen great progress in terms of laser cooling of the nanomechanical oscillator down to its quantum ground state [10], phonon lasing [11,12], heat transport [13], phonon leakage engineering [14], and optomechanically induced transparency [15,16,17]. Because of the dynamical back-action in optomechanical interaction, the phonons can be cooled and amplified by enhanced anti-Stokes scattering and Stokes scattering, respectively. The cooling of the phonons can achieve ultra-low phonon occupancy and approach the quantum ground state. The amplification of phonons can realize phonon lasing and acoustic resonation with high frequency. The phonon lasing normally occurs in a single mechanical mode, as in photon lasing. The optomechanical crystal cavities have been studied theoretically and experimentally for generating GHz and even THz phonons by ultrafast laser excitation through Brillouin scattering [9,18,19,20,21,22,23]. Therefore, the optomechanical systems offer great potential for breaking through the barrier of a GHz nanoresonator with high *Q*-factor at room temperature.

In the present work, we propose a novel cross-shaped nanobeam mechanical resonator operating at 5.69 GHz based on optomechanical interaction. The structure can be fabricated by using a silicon-on-insulator (SOI) wafer, and then the cavity geometry is defined by etching on the silicon wafer with electron beam lithography. Large coupling rates and high *Q*-factor in room temperature are therefore achieved.

## 2. The Mechanical Nanoresonator Model

As periodic structures, optomechanical crystals (OMC) possess the properties of both phononic crystals and photonic crystals [24]. In other words, acoustic and optical band gaps (BGs) can be simultaneously generated in such structures. In practice, for a periodic structure, only a few unit cells are sufficient to attenuate the propagation of acoustic and optical waves. Such a nominal unit cell with both acoustic and optical band gaps (BGs) is a prerequisite for designing an optomechanical crystal cavity. However, these BGs are not necessarily complete BGs. “Quasi-band gaps” for certain specific polarizations also meet the demand.

As shown in Figure 1, a quasi-1D optomechanical nanobeam is proposed as the structure of the mechanical nanoresonator. The nanobeam consists of one cavity and two mirrors. One cavity cell with lattice constant *a*_1_ is placed in the middle of the cavity. As a Bragg reflector, each mirror is composed of multiple normal cells with lattice constant *a*, which is located at each end of the nanobeam. Since the cavity cell and the normal cell have some different geometrical parameters, the other cells in the cavity should meet tapered functions to avoid the dimension discontinuity. Two mirrors reflect back the acoustic and optical waves. The traveling waves are then converted into standing waves. Consequently, acoustic and optical localized modes arise in the small cavity. Owing to the small volume of the modes, the interaction between phonons and photons is strongly boosted [25,26]. 

The optomechanical interaction results from two mechanisms: (1) The photoelastic effect (PE) [27]; permittivity variation induced by strain field; (2) the moving interface effect (MI) [28]; permittivity variation caused by the moving of the interface. The cavity radiation field, i.e., radiation pressure, couples with the mechanical motion. Hence, the fundamental mechanism is the momentum transfer of photons [29]. The enhanced phonons can be excited by a picosecond laser [30]. Consequently, the optomechanical crystal cavity can be utilized as a mechanical resonator owing to the strong coupling between phonons and photons.

To quantify the coupling rates, we considered both the frequency shifts due to the photo-elastic effect [27] and the moving interface [28].
(1)$$g_{\mathrm{OM}}=g_{\mathrm{PE}}+g_{\mathrm{MI}}=-\frac{\omega }{2}\frac{\langle E\left|\Delta \varepsilon_{\mathrm{PE}}\right|E\rangle +\langle E\left|\Delta \varepsilon_{\mathrm{MI}}\right|E\rangle }{\langle E\left|\varepsilon \right|E\rangle }x_{\mathrm{zpf}}$$
(2)$$\langle E\left|\varepsilon \right|E\rangle =\int_{V}\varepsilon {\left|E\right|}^{2}dV$$
where ω is the optical angular frequency. ε is the permittivity matrix of the materials, while ε is the permittivity of an isotropic material without deformation. **E** represents the electric field. $x_{\mathrm{zpf}}=\sqrt{\hbar /\left(2m_{\mathrm{eff}}\Omega \right)}$ is the zero-point fluctuations with the reduced Planck constant ℏ, and the angular frequency Ω of the acoustic mode. $m_{\mathrm{eff}}=\int_{V}\rho {\left|U\right|}^{2}dV$ represents the effective mass with normalized displacement field **U** (max{|U|}=1), The moving boundary contribution is given by mass density ρ. The moving boundary contribution is given by
(3)$$\langle E\left|\Delta \varepsilon_{\mathrm{MI}}\right|E\rangle =-\frac{\omega }{2}\oint_{\partial V}\left(U\cdot n\right)\left(\Delta \varepsilon E_{∥}^{2}-\Delta \varepsilon^{-1}D_{⊥}^{2}\right)dS$$
where **n** is the outward normal vector of the surface ∂V. **D** is the electric displacement field. The subscripts || and ⊥ denote the field component parallel and perpendicular to the interface between two dielectric materials, respectively. $\Delta \varepsilon =\varepsilon_{\mathrm{Si}}-\varepsilon_{\mathrm{air}}$ and $\Delta \varepsilon^{-1}=\varepsilon_{\mathrm{Si}}^{-1}-\varepsilon_{\mathrm{air}}^{-1}$. As for the photoelastic effect contribution [31],
(4)〈E|ΔεPE|E〉=−ε0n4∫{2Re{Ex*Ey}p44Sxy+2Re{Ex*Ez}p44Sxz+2Re{Ey*Ez}p44Syz+|Ex|2[p11Sxx+p12(Syy+Szz)]+|Ey|2[p11Syy+p12(Sxx+Szz)]+|Ez|2[p11Szz+p12(Syy+Sxx)]}dV
where $\varepsilon_{0}$ is the permittivity of free space. *n* is the refractive index. *S_ij_* (*i*, *j* = *x*, *y*, *z*) denote the strain components. Re{} denotes taking a real part. Superscript * represents conjugation. 

## 3. Results and Discussions

The mechanical nanoresonator is designed and optimized to improve a qualify factor as follows. First, the geometry of the nominal cell is determined to provide a desired acoustic fundamental mode. Second, based on the same topology of the nominal cell, the cavity cell is determined by guiding its acoustic and optical fundamental modes into the acoustic and optical band gaps of the nominal cells, respectively. Then, the acoustic and optical fundamental modes can be localized in the cavity. Thirdly, a tapered function is given to make sure that the nanobeam is smoothly transformed from the nominal cell to the cavity cell, and from the ends to the cavity center. Finally, the mechanical losses are investigated comprehensively, and the mechanisms of the high *Q*-factor are revealed.

### 3.1. Design and Optimization

We present a cross-shaped unit cell of a silicon wafer for designing the quasi-1D optomechanical nanobeam, as schematically illustrated in Figure 2a. The geometrical parameters as lattice constant *a* = 600 nm, *b* = 180 nm, *c* = 180 nm, *w* = 600 nm are fixed unless otherwise stated. The thickness *t* is taken as 220 nm, which is commonly used in the SOI platform. The coordinate system is shown in Figure 1. Crystallographic orientation (100) corresponds to the *x*-axis. The acoustic and optical material parameters of Si are: mass density *ρ* = 2330 kg/m^3^; anisotropic elastic constants *C*_11_ = 166 GPa, *C*_12_ = 64 GPa, *C*_44_ = 80 GPa; refractive index *n* = 3.5, and the photoelastic constants *p*_11_ = −0.094, *p*_12_ = 0.017, *p*_44_ = −0.051 [31]. 

First, we classify the modes as even (e) and odd (o) symmetry with respect to the middle planes of the unit cell, as shown in Figure 2a, i.e., the planes of *y* = 0 or *z* = 0. Two letters, such as oe, denote the mode has the odd symmetry about the plane *y* = 0, and even symmetry about the plane *z* = 0. Only the ee symmetry bands are considered since it can prevent the cancellation of the optomechanical interaction caused by superposition between acoustic and optical modes [32]. In order to avoid the cancellation due to the anti-phase motion between adjacent cells, it is most desirable to concentrate the vibration in the *y* direction. Figure 2b shows the band structure of the PnC with an ee symmetry cross-shaped unit cell, where *w* = 500, 600, and 700 nm, respectively. The corresponding mode shapes marked in the band structures are displayed in Figure 2c. For *w =* 600 nm (i.e., *w* = *a*), the displacements in *x* and *y* directions have the same amplitude, and in-phase for the 4th mode, but out-of-phase for the 3rd mode. By changing *w*, the displacements in *x* and *y* directions have different amplitudes. For *w =* 500 nm (i.e., *w* < *a*), the displacements of the two modes concentrate in different directions: the 3rd mode is *x* direction, while the 4th mode is *y* direction. Conversely, for *w =* 700 nm (i.e., *w* > *a*), the directions of displacements concentrated for the 3rd and 4th modes are *y* and *x* directions, respectively. Accordingly, size *w* has great influence on the vibration mode. For *w* = 500 nm, the BG between the 3rd and 4th bands is too narrow to be utilized as an optomechanical crystal cavity. The geometry parameter of the nominal unit cell is chosen as *w* = 700 nm for the following two reasons. First, among the three lengths, the structure of *w* = 700 nm has the widest band gap between the 3rd and 4th bands, which is displayed by the blue region in Figure 2b. Second, compared with the T_1_ and T_2_ bands, the T_3_ band has the smallest slope. This means that its group velocity is also the smallest. Hence, the energy of the wave does not propagate outward. As a result, the mode T_3_ is suitable as a fundamental acoustic mode to form a guide cavity mode.

Figure 3a,b show the acoustic and optical band structures of the nominal unit cell, respectively. It should be pointed out that the optical band structure is obtained by applying an even symmetry boundary condition on the plane of *z* = 0. Periodic boundary conditions are set on the axial boundaries. In addition, the scattering of optical waves in the air should also be considered in electromagnetic simulation. Therefore, an air box is set around the unit cell, except in the axial direction. The scattering boundary conditions are then applied to the boundary of the air box. For acoustic waves, the scattering is negligible due to the huge density difference between silicon and air. The blue shadow zones in band structures showing acoustic and optical BGs are 5.58~6.84 GHz and 180~211 THz, respectively. As shown in Figure 1b, the nanobeam resonator consists of multiple tapered cavity unit cells. The cavity unit cell is further determined according to the nominal unit cell. The nominal unit cell is smoothly transited to the cavity unit cell by adjusting geometry parameters *a_i_*, *b_i_*, *c_i_*, and *w_i_*, as illustrated in Figure 1b and Figure 2a. As the width of the connector between two adjacent unit cells, a same geometrical parameter *b_i_* is taken for different unit cells. To make the modes T_3_ and O_1_ guided modes simultaneously, the acoustic and optical frequency of the cavity unit cell should locate within the acoustic and optical BGs, i.e., 5.58~6.84 GHz and 180~211 THz, respectively. Meanwhile, the optical cavity mode with a frequency of 194 THz is adopted. The optical mode of the resonator operating at a communication wavelength (such as 1550 nm) can be utilized for the signal processing of an electromagnetic wave, which makes it a multifunctional device integrating both acoustic and optical signal processing. 

Figure 3c–e are plotted to show the frequencies of modes T_3_ and O_1_ varying with the geometrical parameters of the cavity cell. The ranges of the geometrical parameters are chosen for the following reasons: (1) To form a small tapered cavity, the lattice constant of the cavity cell is smaller than that of the nominal unit cell. (2) The mode T_3_ is obtained from *w* > *a*; hence, a *w*_1_ larger than *w* is taken as shown in Figure 1. (3) Parameter *c*_1_ can be larger or less than *c*. It can be seen that the frequencies of modes T_3_ and O_1_ have the same trend; both of them decrease as the geometric parameters increase. Additionally, with an increase of *w*_1_, the frequency of mode T_3_ decreases dramatically and far below the acoustic BG. Thus, *w*_1_ = 700 nm (i.e., *w*_1_ = *w*) is fixed in the following design procedure. That is, all cells inside the cavity have the same *w*, *b*, and *t* as nominal unit cells. Only *a_i_* and *c_i_* of the cells need to be adjusted in the cavity.

In order to determine geometrical parameters *a*_1_ and *c*_1_, dependence of the frequency contours of modes T_3_ and O_1_ upon these two geometrical parameters is plotted in Figure 4a,b, respectively. For acoustic mode T_3_, the frequency within the acoustic BG is illustrated by only one yellow triangle in the left bottom, while for optical mode O_1_, the frequency within the optical BG distributes almost the whole left bottom of the square. The acoustic and optical guided modes can be obtained simultaneously by taking the value in the parameter overlapping regions as Figure 4a,b. Correspondingly, frequencies of modes T_3_ and O_1_ are both located in corresponding BGs, such as *a*_1_ = 0.8 *a* and *c*_1_ = 0.7 *c*.

Figure 5 displays the frequencies of the 2nd and 3rd acoustic modes S and T versus *a*_1_, where *c*_1_/*c* = 0.76, 0.94, and 1. Hence, one intersection exists between two frequency lines of these two modes for each *c*_1_. It can be noted from Figure 5 that the frequency of acoustic mode S_3_ increases faster than that of the acoustic mode T_3_ as *a*_1_ decreases from 1 to 0.7. On the frequency of the intersection, a quasi-degenerate mode generates and leads to aberration of the mode shape because of coupling between two modes T_3_ and S_3_. Hence, the size around *a*_1_ = 0.85 *a* of the intersection is urgently avoided. When *a*_1_ decreases across the intersection, modes S_3_ and T_3_ convert to the 3rd and 2nd modes, respectively.

Since the geometrical parameters of the nominal unit cell and the cavity cell have been determined, it is further necessary to determine structural tapered functions of the transformation from the nominal cell to the cavity cells. As shown in (a) and (b) of Figure 6, *a_i_*(*x*) and *c_i_*(*x*) are the structural tapered functions. Because the significant structural difference between the adjacent unit cells leads to large optical scattering losses [33], the boundary conditions ${dp\left(x\right)/dx|}_{x=0,1}=0$ need to be satisfied to reduce the losses. The basic function $p\left(x\right)=2x^{3}-3x^{2}+1$ is defined to construct structural tapered functions as $a_{i}\left(x\right)=a\left[1-\left(1-\frac{a_{1}}{a}\right)p\left(x\right)\right]$ and $c_{i}\left(x\right)=c\left[1-\left(1-\frac{c_{1}}{c}\right)p\left(x\right)\right]$. The variation curves of $a_{i}\left(x\right)/a$ and $c_{i}\left(x\right)/c$ along the nanobeam resonator are plotted in Figure 6a. The geometric parameters are listed in Table 1. Figure 6b,c illustrate optical and acoustic guided modes which are highly localized in the cavity. In the electromagnetic simulation, the whole structure is surrounded by an air box. Since the air box has no periodic boundary, scattering boundary conditions are adopted for all boundaries of the air box. It is worth noting that these two guided modes are exactly transformed from the fundamental modes O_1_ and T_3_. Compared to air, silicon has a mass density of three orders of magnitude greater, while its refractive index is of the same magnitude. The acoustic guided mode is completely localized in the cavity cell while the optical guided electric field is distributed in multiple units. Nevertheless, the optical guided mode yields a *Q*-factor of *Q*_o_ = 2500 by considering the electromagnetic scattering. Then, according to Equations (1)–(4), the coupling rate between these two modes is obtained as (*g*_PE_ + *g*_MI_)/2π = (442 + 333) kHz = 775 kHz.

### 3.2. Mechanical Losses

Until now, much research has been devoted to reduce mechanical losses to improve the quality factor of the mechanical resonator [34,35,36,37,38,39]. The sources of the mechanical losses include the Akhieser mechanism [40], thermoelastic damping [41], and clamping losses [42]. Hence, the total mechanical *Q*-factor *Q_m_* is given by corresponding *Q*-factors [43]:(5)$$\frac{1}{Q_{m}}=\frac{1}{Q_{AK}}+\frac{1}{Q_{TED}}+\frac{1}{Q_{CL}}$$

Obviously, the total *Q*-factor is mainly determined by the $\min\left\{Q_{i}\right\}$ as in the buckets effect. The first term on the right side of Equation (5) can be received as [44]:(6)$$Q_{AK}=\frac{\omega_{m}}{2\alpha c_{s}}$$
(7)$$\alpha =\frac{\gamma^{2}\omega_{m}^{2}C_{V}T\tau }{2\rho c_{s}^{3}}$$
(8)$$\tau =\frac{3\kappa }{C_{V}c_{s}^{2}}$$
where *C_V_* is the volumetric specific heat at constant volume, *γ* is the average Grüneisen coefficient, and *κ* is the thermal conductivity. *c_s_* is the average velocity of longitudinal and shear waves. All of these parameters are temperature dependent. Consider Si at room temperature (300 K), *γ* = 0.4556 [45], *κ* = 155 W/(m·K) [46], *C_V_* = 1.6611 J/(cm^3^·K), and *c_s_* = 9129.9 m/s [47]. Then, substituting these parameters into Equations (6)–(8), the *Q_AK_* is obtained as 1.5 × 10^4^.

For the second term *Q_TED_* of Equation (5), it arises from the coupling between the displacement field and the temperature field [48]. The expansion (and contract) of a structure under cyclic stress causes local cooling (and heating) and leads to temperature gradients. By considering the thermoelastic damping, the normalized thermal profiles *T* of two phases, 0 and π, are shown in Figure 6d,e, respectively. All boundaries are free and have thermal insulation. The *Q_TED_* is further obtained by FEM. The temperature field and the displacement filed are highly overlapped. Moreover, both the temperature and displacement fields are highly confined inside the cavity. Therefore, it is known that the temperature change is caused by high-frequency mechanical vibration. Given that the coefficient of thermal expansion *β* = 2.6 × 10^−6^ K^−1^, the QTED=Re{ωm}/2Im{ωm} is obtained as 5.1 × 10^4^.

As depicted in Figure 7b, the perfectly matched layers (PML) with the length of 2*a* are added to two ends of the nanobeam. The boundaries are all free, and then the logarithm normalized displacement field is obtained in Figure 7c by modal analysis. The displacement amplitude decays exponentially from the cavity to the mirror at two ends. Under the free boundary condition at both ends, the displacement of the end is 10 orders of magnitude smaller than that of the cavity. Considering all symmetry bands, the acoustic band structure of the PnC with nominal cells is depicted in Figure 7a. It can be noted that the acoustic cavity band of 5.69 GHz lies in the complete BG, which greatly reduces the clamping losses and yields a *Q_CL_* as large as 1.2 × 10^9^.

Compared with *Q_AK_*, *Q_TED_* is several times larger, while *Q_CL_* is several orders of magnitude larger. Therefore, from Equation (5), the acoustic quality factor *Q_m_* at room temperature is 1.17 × 10^4^, which is close to *Q_AK_*. Mainly limited by the Akhieser mechanism [40], the *Q*-factor of the proposed nanoresonator cannot be further improved unless through cryogenic processing. From Equations (6)–(8), it can be found that the *Q_AK_* is inversely proportional to frequency. Consequently, a further increase of frequency of the resonator will inevitably reduce the *Q*-factor. Nevertheless, the *f* × *Q_m_* product is as large as 8.5 × 10^13^, which is considerable at room temperature and GHz resonance. Finally, Table 2 lists the performance comparison among different types of the resonators in published literature and this work. The proposed resonator has the highest frequency and the largest *Q*-factor at room temperature. The resonator can even reach a *Q*-factor as large as 4.9 × 10^10^ by operating in a dilution refrigerator at milliKelvin temperature [37].

## 4. Conclusions

A novel cross-shaped resonator with an optomechanical crystal cavity was proposed to break the low quality bottleneck factor at a high frequency GHz level. The design idea and optimized procedure were provided systematically. An acoustic guided mode highly localized in the cavity was obtained by utilizing a fundamental mode with transverse vibration. Meanwhile, inside the cavity, the high overlap between optical and acoustic modes enhanced the coupling rates. The clamping losses were greatly reduced by complete BG of the mirrors. By taking a thorough consideration of all kinds of mechanical losses, we found that the acoustic *Q*-factor is mainly limited by the Akhieser mechanism at room temperature. Finally, the nanoresonator was realized, which can reach *Q_m_* = 1.5 × 10^4^ and the coupling rate of 775 kHz even when operating at 5.69 GHz. Therefore, when operating at a GHz range, the proposed nanoresonator can produce a *Q*-factor two orders of magnitude larger than SAW and FABR. This work paves the way for designing a GHz nanoresonator with a high *Q*-factor.

## Figures and Tables

**Figure 1 micromachines-13-01862-f001:**
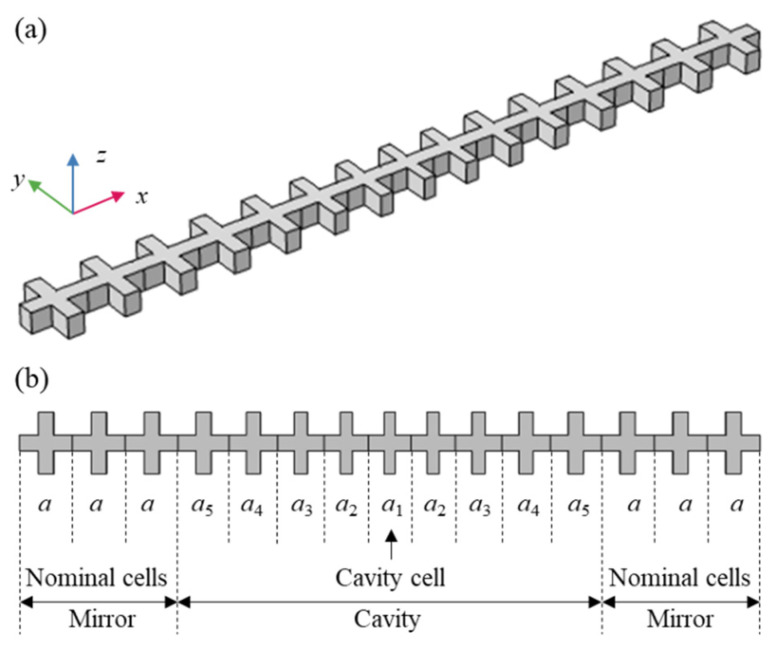
The mechanical nanoresonator. (**a**) The 3D schematic diagram; (**b**) the view of the *x-y* plane.

**Figure 2 micromachines-13-01862-f002:**
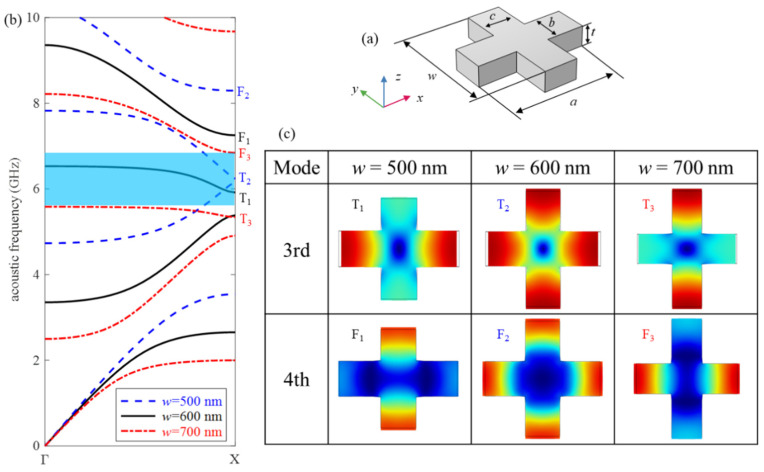
(**a**) A cross-shaped unit cell. (**b**) Acoustic band structure of ee symmetry modes for *w* = 500, 600, and 700 nm, respectively. (**c**) Mode shapes of the 3rd (denoted by T) and 4th (denoted by F) modes for different *w*.

**Figure 3 micromachines-13-01862-f003:**
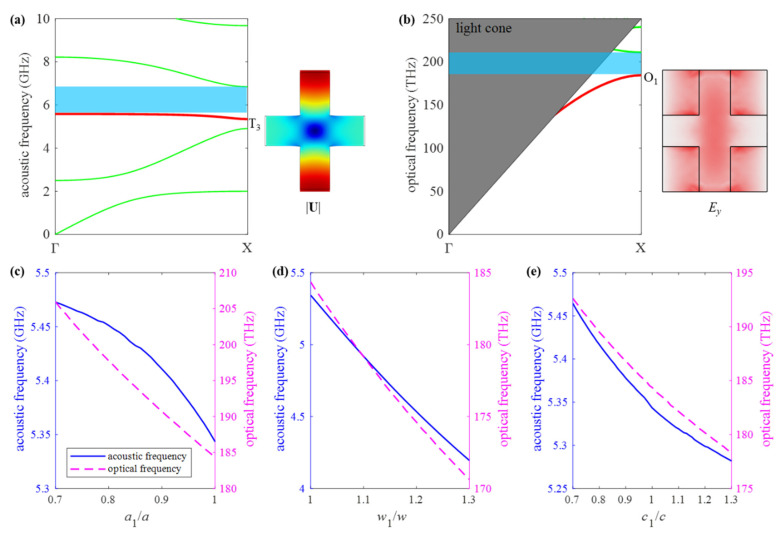
(**a**) With a nominal unit cell, dispersion curve of (**a**) the PnC, and ee symmetric displacement field of the mode T_3_; (**b**) the PtC with ee symmetric electric field of the mode O_1_. The frequencies of the modes T_3_ and O_1_ vary with the geometrical parameters; (**c**) *a*_1_, (**d**) *w*_1_, (**e**) *c*_1_.

**Figure 4 micromachines-13-01862-f004:**
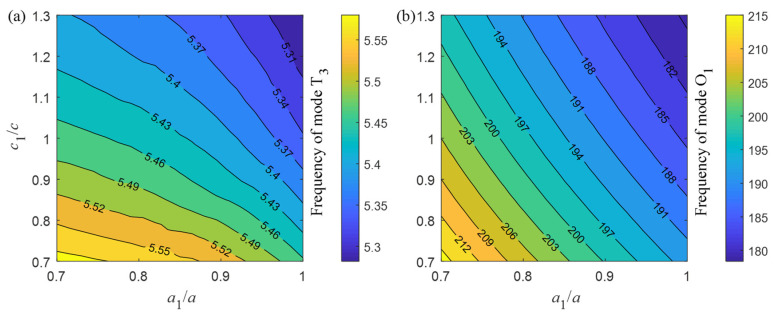
Frequency contours varying with geometrical parameters *a*_1_ and *c*_1_ for (**a**) mode T_3_, (**b**) mode O_1_.

**Figure 5 micromachines-13-01862-f005:**
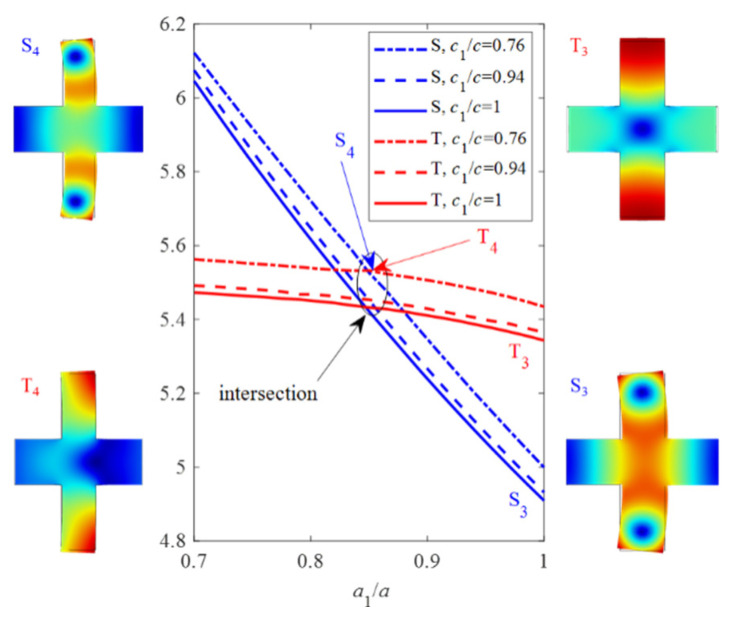
Frequency shifts of the 2nd and 3rd modes as *a*_1_ varies, and the vibration shape at specific points.

**Figure 6 micromachines-13-01862-f006:**
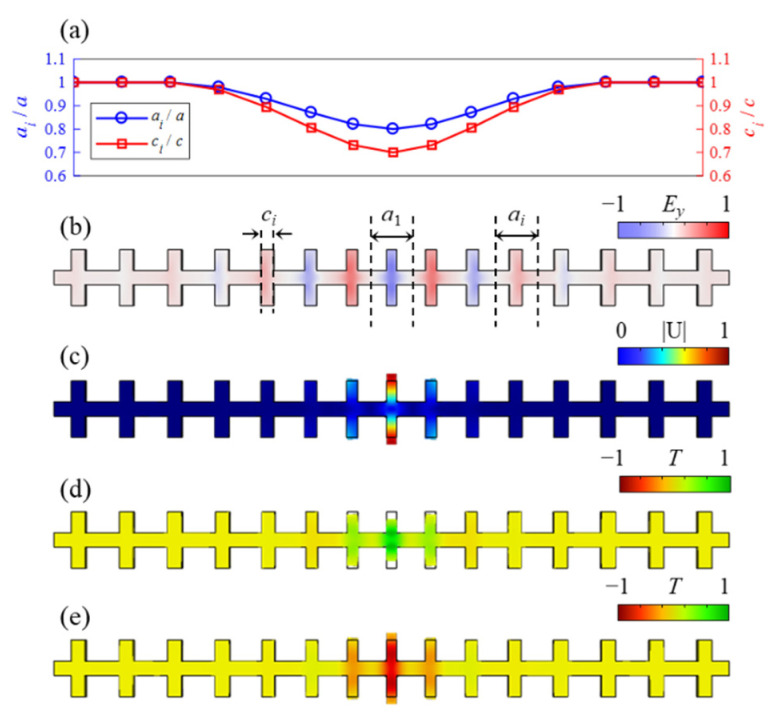
(**a**) Structural tapered function curves *a_i_*(*x*) and *c_i_*(*x*). (**b**) The normalized electric field *E_y_* for the optical guided mode at 198 THz. (**c**) The normalized displacement field |**U**| for the acoustic guided mode at 5.69 GHz. The normalized thermal profile *T* of two phases: (**d**) 0, and (**e**) π, which is solved by the thermomechanical FEM for the acoustic mode at 5.69 GHz and room temperature.

**Figure 7 micromachines-13-01862-f007:**
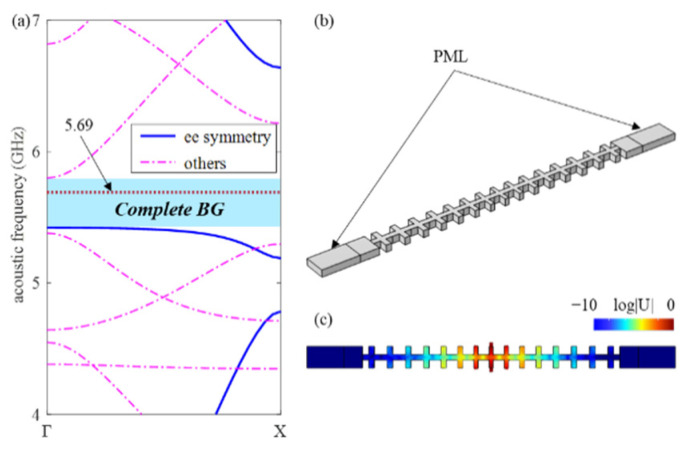
(**a**) The complete band structure of the PnC with nominal unit cells. (**b**) The model to calculate clamping losses. (**c**) Logarithm normalized displacement field of acoustic cavity mode at 5.69 GHz.

**Table 1 micromachines-13-01862-t001:** Varied geometric parameters of the cavity cells. Note that the other geometric parameters *b*, *w*, and *t* of the cavity cells are all fixed.

Cell Number *i*	1	2	3	4	5
*a_i_* (nm)	480	492	522	558	588
*c_i_* (nm)	126	132	145	161	174

**Table 2 micromachines-13-01862-t002:** A performance comparison among different types of resonators operating at GHz.

Resonator Type	Material	Resonant Frequency (GHz)	*Q*-Factor	*f.Q_m_*
SAW [6]	AlN	1.325	109	1.4 × 10^11^
FBAR [7]	PI	1.055	210	2.2 × 10^11^
FBAR [49]	AlN	2.5	850	2.5 × 10^12^
OMC Resonator [37]	Si	5	4.9 × 10^10^	2.6 × 10^20^
This work	Si	5.69	1.17 × 10^4^	8.5 × 10^13^

## Data Availability

Data underlying the results presented in this paper are not publicly available at this time but may be obtained from the authors upon reasonable request.

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
