# Peer review of "Design of GHz Mechanical Nanoresonator with High Q-Factor Based on Optomechanical System"

_micromachines, 2022, doi:10.3390/mi13111862_

Round 1
Reviewer 1 Report
The authora s have proposed MEMS-based -device on optomechanical interactions to overcome the shortcoming of PZT. The resonance can be excited by a picosecond laser through the radiation pressure. The manuscript is well written and fits the general scope of the journal. Below are some concerns regarding this submission:
- The introduction section is not enough. The authors have to describe the previous related work not only cite them.
- Figure 1 shows that the structure is symmetric. What would happen if it is asymmetric. Will the Q change?
- A table listing the device parameters and dimensions needs to be added.
- Why the authors have ignored the first two resonant modes?
- To excite the higher modes, we need high power because they are stiffer, have the authors noted this?
- A sensitivity analysis needs to be conducted to show how the quality factor can change as the main parameters changes.
- From a practical point of view, which fabrication process can be used to manufacture the device?
- Most of the analysis is analytical, would it be possible to validate your concept with other related experiments?
Thanks
Reviewer 2 Report
The authors proposed a high Q-factor nanoresonator at GHz by utilizing optomechnical nanobeam consisting of silicon cross units, and numerically calculated the band gaps, displacement modes and Q-factor. This work is complete and interesting. However, there are some issues should be solved before the acceptance.
1. The band gap of the proposed structure is 5.58GHz-6.84GHz and such a range should be called as microwave frequency band. However, the authors regarded this as acoustic band. The microwaves are transverse waves and the acoustic waves can be transverse or longitudinal waves, and the two types of waves should be classified. Please confirm the term of the waves ranging from 5.58GHz to 6.84GHz or explain the so-called acoustic waves defined in their work.
2. The authors should check the English expressions in the total manuscript. For example, the ‘consider’ should be corrected as ‘considered’ on page 3, 99th row.
3. In Figure 4, the caption is hard to understand for readers. Please identify that the Figure 4(a)-(b) depict frequencies of T3 and O1 mode. The colormap is defined as frequency, but the colorbar disappears.
4. In the manuscript, the simulation settings should be illustrated such as boundary conditions, excitation sources, etc.
5. For Table 1, the Q-value are seemed as very high compared with previous work. Maybe, the authors should also refer some similar work (10.1109/TMTT.2016.2607712, High Q-factor Silicon Resonator for High Frequency Oscillators, etc).
Round 2
Reviewer 1 Report
Thanks for the updated version. If is in a good shape now.
Reviewer 2 Report
the authors have addressed all my previous concerns, and the work can be published along with other reviewer's comment.